# Crack Length of Elastomeric Sealants and Their Service Life in Contrasting Canadian Climates: Effects of Climate Change

**DOI:** 10.3390/polym16142039

**Published:** 2024-07-17

**Authors:** Marzieh Riahinezhad, Elnaz Esmizadeh, Itzel Lopez-Carreon, Abhishek Gaur, Henry Lu, Michael A. Lacasse

**Affiliations:** Facade Systems and Products Group, Construction Research Centre, National Research Council Canada, 1200 Montreal Rd, Ottawa, ON K1A 0R6, Canada; marzieh.riahinezhad@nrc-cnrc.gc.ca (M.R.); itzel.lopez-carreon@nrc-cnrc.gc.ca (I.L.-C.); abhishek.gaur@nrc-cnrc.gc.ca (A.G.); henry.lu@nrc-cnrc.gc.ca (H.L.); michael.lacasse@nrc-cnrc.gc.ca (M.A.L.)

**Keywords:** service life, silicone sealant, crack growth, climate stressors, cyclic movement

## Abstract

The longevity of polymer-based sealant and jointing products, including elastomers, significantly depends on the level of exposure to sunlight and joint movement. These factors are particularly crucial in the application of polymers in construction due to their susceptibility to degradation under environmental conditions. For instance, diurnal cycles of contraction and dilation, arising from daily temperature fluctuations, impose significant stress on sealants and joints, impacting their durability over time. The elastic nature of polymeric sealants enables them to endure these cyclic mechanical loads. Athough there is considerable information on sealant durability obtained from laboratory accelerated aging, there is limited knowledge about the effect of climatic factors using historical and projected weather data on the durability and expected service life of these products. This study employed the Shephard crack growth model to predict the performance of sealants in a Canadian context; the crack growth and time-to-failure of hypothetical silicone sealants were investigated across 564 locations, for which historical climate data were obtained from 1998 to 2017, including gridded reanalysis data for the period of 1836–2015. The historical climate data were classified into four climate categories, and crack growth was estimated based on historical climatic data within the valid range for the Shephard model, revealing that locations in colder climates with lower levels of precipitation typically exhibit higher cumulative crack growth. The impact of climatic variation and environmental stressors on the longevity of sealants in the context of climate change was also investigated using future projected data.

## 1. Introduction

Cracking of building envelope components due to fatigue is a problem that naturally occurs, given that envelope components are frequently subjected to variable environmental loadings such as fluctuations in temperature, humidity levels, and intermittent exposure to ultraviolet (UV) radiation [1,2]. This concern extends to sealants that are used to seal expansion/contraction joints in the building envelope [3]. Joint and sealant products play a crucial role in maintaining the weather tightness of buildings by serving as integral components within both air and water vapor barrier systems [4]. Given the nature of their placement and intended function, sealants must be sufficiently flexible to accommodate thermally induced joint movement through daily, seasonal, and annual temperature fluctuations. Sealants must be able to expand and contract in response to cyclic fatigue loading caused by temperature cycles [5]. Beyond the requirement for durability against cyclic loading, sealants must also be resistant to environmental aging factors. These include solar radiation, moisture in the forms of condensation, humidity, and rainfall, wind loads, large temperature fluctuations, and physical wear and exposure to chemicals during façade-cleaning [6,7]. In areas such as Canada and similar regions with a cold climate, sealants frequently suffer premature failure, reflecting the extreme conditions to which they are subjected over their service life [8,9]. The impact of these aging factors on the material properties of the sealant determines its long-term durability [10]. Previous studies across various sealant classes, including polyurethanes [11], polysulfides [12], and silicones [13], demonstrated that cyclic movement stands out as the leading factor contributing to degradation, above environmental stresses alone. Moreover, when cyclic fatigue is coupled with elevated temperatures and high relative humidity (RH), this can result in synergistic effects that considerably accelerate sealant degradation [10].

Sealant failure can cause undue consequences such as air and moisture penetration across the building envelope, which can lead to a decrease in energy efficiency and degradation of other building envelope materials and components [14]. Understanding the performance of sealants is crucial to ensure their effectiveness over their intended service life, especially taking into account the future climate and the effects of climate change [15,16]. To address these concerns, degradation and service life models are essential for determining maintenance and replacement schedules, as well as for predicting how sealants will perform in different climates [17]. Research in the construction sector has revealed a significant failure rate for sealants, with 50% failing within 10 years of installation and 95% failing within 20 years. This underscores the crucial need for developing reliable numerical models for fatigue cracking prediction [16]. A limited number of studies have focused on predicting the degradation of sealants from accelerated aging and outdoor exposure experiments, and these have revealed that different types of sealants exhibit varying failure mechanisms in response to identical exposure conditions [18]. Furthermore, the durability of a sealant can be significantly influenced by the service location due to disparities in environmental factors such as UV radiation, duration of wetness, pollutants, mean annual temperature, and other factors that degrade sealant products [19]. One of the primary indicators that can be used to monitor sealant performance is the rate of crack growth since this can directly impact the effectiveness and longevity of sealant products used in construction applications. Studying crack growth in sealants involves analyzing the mechanisms, rates, and patterns of crack propagation, all of which are influenced by environmental factors [20]. Researchers have often employed mathematical models, laboratory testing, and real-world observations to better comprehend and predict crack growth behavior in building materials [21].

Degradation models for sealant materials have primarily focused on two types of deterioration mechanisms: rupture within the bulk of the sealant, classified as cohesive failure [22,23,24], and loss of adhesion to the substrate, classified as adhesive failure [25]. While few studies have used these models to estimate the service life of jointing systems, none have specifically investigated the influence of seasonal fluctuations and climate change. This presents a literature gap in utilizing existing models to enhance our understanding of sealant performance under diverse weathering conditions in Canada, taking climate change into account.

This study introduces a novel approach in which historical data from multiple locations across Canada are applied to an empirical crack growth model to investigate sealant crack progression over time. The impact of climate change is also considered using projected future climate data for selected locations. By incorporating historical and projected future data, a fresh dimension has been introduced to research on the long-term performance of sealant products by capturing the cumulative impact of various environmental stressors over extended periods.

### 1.1. Theoretical Background

While fatigue cracking is the predominant failure mode in sealants, there is still a lack of predictive models for crack growth rate and a comprehensive understanding of this phenomenon. The most notable endeavor to establish a connection between crack-growth rate and applied stresses was initiated by Paris et al. in 1961, introducing the well-known Paris power law [26]. Despite numerous investigations, there is still a pressing need to develop a reliable mechanistic crack growth model for sealants. Following this effort, another sealant crack growth model was developed by Shephard et al. in 1995, using laboratory data from silicone sealants that underwent accelerated aging [27]. The Shephard crack growth model is recognized for its application in investigating how cracks in sealant products behave under cyclic loading. This model was used to predict crack growth rate in sealants located in four contrasting climates in North America: Phoenix, Miami, Ottawa, and Winnipeg. 

This model relates specifically to a butt-jointed aluminum panel system [27] (see Figure 1), which experiences cyclic extension and contraction corresponding to the diurnal, seasonal, and annual temperature fluctuations over the service life of the sealant. 

When temperatures fall, the joint between panel substrate opens, causing the expansion of the sealant bead. Conversely, when temperatures rise, the panels on either side of the joint expand, decreasing the joint gap and subjecting the sealant bead to compression. To maintain balance over the year between periods of tension and compression, it is common practice to apply sealants when the ambient temperature closely aligns with the annual average temperature for the service. This approach minimizes the risk of extreme compression or extension when sealants are applied at temperatures significantly lower or higher than the yearly average, respectively. In the Shephard model, the crack growth rate referring to cracks developing at the interface between a silicone sealant and an aluminum substrate can be expressed by Equation (1).
(1)a˙=(Gk)1n.1aT.aRH,
where $\dot{a}$ is the crack growth rate (m/s), k is a constant with a value of 1161.7, *n* is a constant with a value of 0.184, $a_{T}$ is the temperature (*T*)-dependent shift factor ($\log a_{T}=6841T-20.81$), $a_{RH}$ is the relative humidity (*RH*)-dependent shift factor ($\log a_{RH}=9.253-0.266RH+0.0016{RH}^{2}$), and G is the strain energy release rate, or tearing energy, that can be calculated by Equation (2) [28].
(2)$$G=\int_{x_{0}}^{x}\left(\left(236040T+1.07e^{8})x+(-437T+24827\right)\right)dx,$$
in which *x* is the displacement of the joint (x=1.7.a.T), and *a* is the thermal linear expansion coefficient of aluminum.

In the current study, the Shephard crack growth model was systematically employed across a range of climatic scenarios in Canada, utilizing both historical and predicted future climate data. The aim was to develop a thorough understanding of the anticipated performance of sealant products over time, capturing the diverse and dynamic range of weather conditions that these joint systems might encounter. In addition, projected future climate data were incorporated at select locations to investigate the implications of climate change. A uniform installation temperature of 30 °C was assumed across all locations;the reasoning for this decision is described in detail in Section 2.2. Joint failure was defined to occur when the crack length exceeded 100 mm [27].

### 1.2. Study Area and Climate Data

This study was performed in Canada, a country inhabited by more than 38 million people who reside and work in over 15 million residential buildings and over 480,000 commercial and institutional buildings [29]. Canada’s climate exhibits significant variation across its expansive territory. A typical north-to-south temperature gradient is evident; the average summertime temperatures vary from approximately 0 °C in the northernmost regions to around 22 °C in the southernmost regions. Notably, the oceans on the West and the East coasts play a pivotal role in regulating regional temperatures, resulting in moderate temperatures in coastal areas. In contrast, the inland provinces such as Alberta, Saskatchewan, and Manitoba (the Prairies) experience relatively extreme winters and summers [30].

Precipitation patterns across Canada also vary significantly, with distinct regional disparities. The western coastal regions of British Columbia receive the highest annual precipitation level, often exceeding 1000 mm; this can be attributed to the convergence of moisture-laden air masses from the Pacific Ocean, coupled with the geographical barrier of the Rocky Mountains. Moving to the Atlantic coastline, provinces such as Newfoundland and Labrador and New Brunswick receive the second-highest levels of annual precipitation in the country, typically exceeding 1000 mm. The precipitation in these areas is primarily cyclic and evenly distributed across the year. The provinces of Ontario and Quebec receive substantial precipitation, falling within the range of 500–1000 mm. This can be due to the presence of significant moisture sources, including the Great Lakes, Hudson Bay, the Atlantic Ocean, and the Gulf of Mexico. Finally, the Prairies and the northernmost regions of Canada receive the lowest amounts of annual precipitation, ranging up to 400 mm. The air masses in these regions tend to be excessively cold and dry, with small amounts of hard and dry snowfalls that are compacted by high winds. Spring and summer are usually wetter than the winter months [30].

In this study, four different types of climate datasets were utilized for different portions of the analysis [31]. The first climate dataset, comprising temperature and relative humidity data measured at 564 monitoring locations in Canada, was gathered from the Canadian Weather Energy and Engineering Datasets (CWEEDS) developed by Environment and Climate Change Canada [32]. The geographical distribution of the monitoring locations is illustrated in Figure 1. The CWEEDS database provides continuous hourly time-series data for a number of climate variables, covering at least 10 years of climate data for the period from 1998 to 2017.

For the second climate dataset, estimates of historical weather data spanning 1836–2015 were extracted from the Twentieth Century Reanalysis Project (version 3) [34]. The reanalysis datasets were produced by the Earth System Research Laboratory (ESRL) under the National Oceanic and Atmospheric Administration (NOAA) and the Cooperative Institute for Research in Environmental Sciences (CIRES) at the University of Colorado. This project leveraged the computational power of the Department of Energy supercomputers to produce a comprehensive global reanalysis dataset covering an extensive timespan from 1836 to 2015. The effort involved assimilating surface observations of synoptic pressure into an 80-member ensemble of estimates for the Earth system. Boundary conditions of mean pentad (five days) sea surface temperature, monthly sea ice concentration, and time-varying solar, volcanic, and carbon dioxide radiative forcing are prescribed in the datasets. In the present study, 3-hourly estimates of temperature and relative humidity were extracted from this reanalysis dataset for the period of 1836 to 2015 for each of the aforementioned 564 CWEEDS locations. The 3-hourly temperature and relative humidity estimates were then interpolated to obtain hourly estimates, and these points were employed to calculate the long-term retrospective crack growth rates.

The third dataset employed in our study comprised the Canadian Climate Normals [35], which represent 30-year averages to succinctly define and characterize the average climatic conditions of specific locations across Canada. Environment and Climate Change Canada periodically revise these climate normals with updates occurring at the end of each decade to ensure that the information reflects the most recent and relevant climatic trends in the regions analyzed. 

The fourth climate dataset in this work was the future projected climate dataset that was prepared for 564 locations widely distributed across Canada (Gaur et al., 2022 [33]). The dataset was prepared by bias-correcting regional climate model projections from the Canadian Regional Climate Model, CanRCM4 [33]. The data provide hourly estimates of several climate variables useful for building simulations, including temperature and relative humidity, under different levels of global warming ranging from 0.5 °C to 3.5 °C. A notable feature of the future projected climate dataset is that the climate files are derived directly from regional climate model projections, allowing it to capture complex distributional changes in future climate [33].

## 2. Analysis and Results

### 2.1. Mapping and Categorization of the Climate Regions

The analysis began with mapping and categorizing the climate of the 564 CWEEDS locations for the purpose of selecting a smaller representative set of locations for further detailed examination. To achieve this, a scatter plot was generated based on the hourly observational climatic data, using temperature and relative humidity data available from 2007 to 2017. The results of this analysis are presented in Figure 2.

Based on the information gleaned from the scatter plot, 18 locations with the most contrasting climatic characteristics were selected. These specific locations are identified in Figure 3 with different symbols. Subsequently, climate data pertinent to these selected locations were gathered from the Canadian Climate Normals website [35] for the time span of 1951–1980. This included temperature, rainfall, snowfall, and total precipitation data, which are comprehensively presented in Table 1. 

The 18 locations were classified into four climate categories: cold/wet, cold/dry, warm/wet, and warm/dry, based on average annual temperature and average annual rainfall or total precipitation (see Table 2):Wet: Average annual rainfall > 500 mm;Dry: Average annual rainfall < 500 mm;Warm: Average annual temperature ≥ 5 °C;Cold: Average annual temperature < 5 °C.

These 18 locations are shown on Canada’s map in Figure 3 to visualize the geographic distribution of these contrasting climates.

### 2.2. Significance of Installation Temperature on Crack Growth

Temperature differentials play a crucial role in crack propagation, which highlights the impact of the installation temperature and temperature variation. When the ambient temperature drops below the installation temperature, the sealant expands due to the contraction of adjacent panels. This widening induces tensile stress within the sealant, which intensifies crack growth. This effect becomes more prominent as the temperature differentials between installation and ambient conditions increase. Therefore, for our initial analysis, it was assumed that sealants were installed at each city’s average annual temperature to simulate a realistic scenario reflective of recommended application conditions. Crack growth on an hourly basis was calculated using historical climate data using the NCEP 20th century reanalysis dataset from 1836 to 2015, within the range applicable in the Shephard model [27]; the results of this analysis are shown in Figure 4.

The findings indicate that the most significant crack lengths developed in cold–dry locations with the highest mean annual temperature ranges, such as Old Crow and Churchill. Conversely, warm–wet locations with lower mean annual temperature ranges exhibited the smallest crack growth over time. Historically, joint failure was assumed to occur when crack lengths exceeded 100 mm in a 2 m joint [27]. However, none of the sealants installed at the 18 selected locations experienced failure, per this definition, which is unrealistic as it is not likely for sealants to last over 100 years in real-field applications. This service-life overestimation can be attributed to the model’s underlying assumptions, specifically that the butt joint is located in a position that is sheltered from direct sunlight and rain. Thus, the model fails to account for some of the various environmental stressors that accelerate sealant deterioration over time. These factors encompass but are not limited to UV exposure, chemical exposure, mechanical stresses, poor installation practices, water ingress, temperature extremes, and biological growth [36,37]. 

Given the limitations of the employed model, the most impactful adjustment to observe accelerated simulated crack growth is to increase the installation temperature. This magnifies the induced tensile stress on the sealants from the contraction of the adjacent panels when temperatures fall below the installation temperature. Figure 5 illustrates the evolution of the accelerated crack growth over time in various locations when the sealants are installed at 20 °C and 30 °C, temperatures commonly experienced in summer in the most populous cities of Canada. In Figure 5, the cumulative crack growth rate over the period from 1836 to 2015 is presented, with the assumption that no failure occurred once the crack length exceeded 100 mm and that there were no restrictions on crack length or material dimensions. The intention for this hypothetical scenario was to explore the long-term effect of time on crack length across different climates. The most notable effects were observed in locations where the disparity between the average annual temperatures and the installation temperature was greatest, as expected. This is particularly pertinent in locations with colder climates, such as Old Crow, where temperature differentials are more pronounced. Furthermore, the range in temperature variation in a specific installation location plays a pivotal role in determining the extent of crack propagation in the jointing product. 

As the intention of this study was to explore the effect of seasonal climatic variations and climate change on the time-to-failure for silicone sealants, an installation temperature of 30 °C was used thereafter for additional detailed analysis. This ensured consistency in observing sealant failure for the vast majority of the selected locations within the timeframe of available climatic data.

The relation between the cumulative crack length and each of the key climate factors, including mean annual temperature, rainfall, and total precipitation, is depicted in Figure 6.

The prevailing trend indicates that colder climates with lower precipitation levels typically exhibit higher cumulative crack growth compared to warmer climates with higher levels of precipitation. For example, the cumulative crack growth results for Alert and Eureka are very similar; both locations show the most substantial crack length over time. Notably, these two locations feature the lowest mean daily, mean minimum, and mean maximum temperatures compared to the other sites. In other words, they boast the coldest climates among the 18 locations studied. Additionally, with regard to precipitation, both Alert and Eureka record the lowest levels of rainfall, snowfall, and total precipitation, making them the driest climates among the ones that were studied. On the other hand, Estevan Point has the lowest crack growth among all the examined locations. This location experiences the highest annual temperature and receives the highest amount of precipitation. This trend is consistent with the crack growth results for the four different locations with contrasting climates, as investigated by Shephard et al. [27].

Installing sealants at temperatures below the mean annual temperature will cause them to undergo greater compressive strain than tension throughout the year. This occurs because, at lower temperatures, the gap between substrates is wider than the mean due to the substrates contracting away from each other; therefore, for the remainder of the year, when ambient temperatures are above the installation temperature, the gap is narrower, and the sealants experience compression. Conversely, installing sealants at temperatures well above the mean, i.e., summer in Canada, will result in the sealants experiencing greater strain in tension as compared to compression throughout the year. In this scenario, the peak extension occurs during the winter when the sealant itself becomes less elastic due to colder temperatures. Consequently, sealants are more likely to fail when installed at temperatures well above the mean annual temperature, which is more obvious in the context of Alert and Eureka [38].

### 2.3. Time to Failure Analysis

The assessment of sealant durability, particularly in diverse climatic conditions, necessitated the integration of historical climate data into the current theoretical model. During this investigation, historical climate data from the selected cities were fed into the Shephard model to evaluate the time-to-failure of the hypothetical sealant with an empirical relation. Time-to-failure can be defined as the period between the installation of the sealant and the point at which the damage reaches its critical threshold, which is defined in this study as a 100 mm crack length. 

To investigate the effect of different locations on the behavior of the hypothetical sealants, historical climate data from the year 1836 were incorporated into the model for various locations that were chosen based on their contrasting climates, as described earlier. Figure 7 illustrates the crack growth over time until the critical length of 100 mm is achieved, signaling the point of failure. Notably, Estevan Point offers a comprehensive view of the entire crack growth process with few gaps, whereas there are significant data gaps for other cities. These gaps arise from instances where the climatic parameters, as used in the model, fell outside the applicable model limits, thereby impacting the overall crack growth data for those particular locations. These gaps serve as valuable indicators, signaling the need for further investigation into the unique environmental stressors experienced at each location and the need to refine the Shephard model to better capture the diverse range of climatic parameters that may occur in different climate zones.

Figure 8 provides a graphical representation of the time-to-failure for hypothetical sealants that were installed in different cities in the year 1836. This figure helps illustrate how long it took for these sealants to reach the point of failure in various locations, shedding light on the durability and performance variations of the sealants under different environmental conditions and stressors across these different locations in Canada. 

Notably, the data highlight that silicone sealants in Estevan Point, characterized by the warmest and the wettest climate, exhibit impressive resilience, with a time-to-failure of approximately 17,000 h (almost 2 years). In comparison, sealants in other locations fail before 7500 h, making the performance in Estevan Point significantly superior. Although a 2-year lifespan was observed compared to the typical life expectancy of sealants in service, it is important to note that failure was deliberately accelerated by selecting an installation temperature significantly higher than the annual mean temperatures. With an installation temperature of 30 °C, the sealants experienced varying degrees of extension when ambient temperatures fell below 30 °C, which was the majority of the time for the selected locations, and extreme compression in the coldest days of winter.

In addition, it is important to acknowledge the presence of missing data for certain cities, particularly those with extremely cold temperatures. This absence stems from the constraints of the Shephard model; due to the narrow temperature range over which it is valid, the climate data for a significant portion of the year during the colder months could not be applied to the crack growth calculations. Despite these gaps, the information obtained provides valuable insights into the performance of sealants in different climates. Among the remaining cities, Bow Island and Bow Valley, with similar mean temperatures and precipitation ranges, manifest the shortest time-to-failure, indicating a comparatively more rapid deterioration of sealant integrity in these locations. The markedly shorter time-to-failure in these cities may indicate more aggressive environmental conditions contributing to accelerated sealant degradation. It is worth noting that the cities lacking information on crack growth, such as Alert and Eureka, happen to experience more severe climates. The absence of data for these locations introduces a significant limitation in our understanding of sealant performance under extreme environmental conditions. This underscores the importance of developing a comprehensive model to cover a more diverse range of climates to ensure a more nuanced and complete analysis of sealant behavior and durability. 

The evolution of time-to-failure versus mean temperature and relative humidity is presented in Figure 9a and b, respectively. The overall trend of the results suggests that sealants installed at 30 °C in cities with lower mean temperatures take less time to reach failure (Figure 9a). The decreased lifespan of the sealant specimens in regions with lower mean temperatures may be explained by the higher extensive stress exerted on these sealants, which is caused by the greater contraction of adjacent panels in such an environment. This, in turn, leads to greater microcrack density and length in the sealants, resulting in an acceleration of macrocrack propagation. It is noteworthy to highlight that, in locations such as Alert and Eureka, where the assumed installation temperature significantly exceeds the annual mean, sealants may undergo more extreme extension, creating a disproportionate impact compared to other regions. Although the relationship between time-to-failure and the mean relative humidity of the installation location is not as evident as the relationship with temperature, it can be inferred from the data trend that the sealants tend to fail sooner when the relative humidity Is higher. The non-monotonic dependency observed in Figure 9b might signal that the combined influence of both humidity and temperature should be considered for the time-to-failure analysis due to the synergistic effect of these two climatic factors.

Crack growth and time-to-failure can fluctuate significantly from one year to the next due to the nature of the Shephard model [27], which depends heavily on specific environmental stressors that vary annually. To illustrate this concept, Figure 10 shows how crack growth in a sealant installed in Estevan Point at the beginning of each decade, from 1836 to 2006, progresses until it reaches the failure point of 100 mm. Estevan Point was selected for its warm and wet climate and slower crack growth rate, which resulted in a greater amount of data to clearly show the trend across different decades. This visual representation illustrates how cracks in the sealant develop over time, offering insights into the different crack growth rates (slope of the curves) during each decade. The model initiates its data collection at 00:00 on January 1^st^ of every decade. The hourly crack growth data reveal discernible patterns, notably with steeper curves indicating a higher crack growth rate during colder months. Subsequently, as spring unfolds, the slope of the curves decreases. The rate at which cracks develop and propagate within the sealant material is directly affected by the unique temperature and humidity conditions prevalent each year, which leads to varying crack growth rates observed from one decade to the next in Figure 10.

The time-to-failure data collected for sealants installed in Estevan Point during different decades are depicted in Figure 11. There is a noticeable variation in the time it takes for sealants to fail, contingent upon the year of installation. This variability can be attributed to the dynamic interplay of environmental factors. The figure reveals that the longevity of sealants, as evidenced by their failure and crack growth rates, is deeply influenced by environmental stressors. The observed increase in the time-to-failure of sealants with increasing installation years, indicating a prolonged resistance before reaching critical crack length, aligns with changes in Earth’s climate since the mid-20^th^ century [39]. This shift in climate is predominantly attributed to the rapid increase in atmospheric carbon dioxide levels since the Industrial Revolution directly linked to human activities, particularly the burning of fossil fuels [40]. The resulting increase in heat-trapping greenhouse gas levels in the Earth’s atmosphere has led to an overall increase in the planet’s average surface temperature. The rate of climate change since the Industrial Revolution has raised global temperatures by nearly 1 °C. This seemingly modest increase can lead to significant global shifts in weather patterns, water cycles, and ecosystems, highlighting the potentially profound consequences of human-induced climate change [41]. However, warmer climate conditions may directly contribute to the enhanced durability of sealants, as evidenced by the lengthened time-to-failure. This connection underscores the intricate relationship between human-induced climate change and the performance of materials in real-world applications.

### 2.4. Seasonal Crack Growth Patterns

The seasonal progression of crack growth in the hypothetical sealant installed at Estevan Point in 1836 is illustrated in Figure 12. This investigation is intended to analyze how environmental factors and weather conditions in different seasons may have influenced the performance of the sealant over time. The results obtained from the study reveal a noteworthy disparity in the sealant crack growth rates between cold and hot seasons, providing valuable insights into how temperature variations affect the sealant’s structural integrity.

Table 3 presents the slopes and coefficient of determination (R^2^) values obtained from a linear regression analysis, depicting the relationship between crack growth and time in the context of Estevan Point. Figure 13 illustrates the seasonal variations in temperature and relative humidity impacting the crack growth of the sealant installed at Estevan Point in 1836, ultimately leading to sealant failure. The results indicate a more pronounced crack growth rate during cold seasons compared to hot seasons. For instance, during the first winter after the onset of crack growth, the average crack growth rate of the sealant, as evidenced by the calculated value of 1 × 10^−5^ mm·h^−1^, is substantially higher than the average crack growth rate of 2 × 10^−6^ mm·h^−1^ calculated for the first summer after the onset of crack growth. The heightened crack growth rate during colder periods can be attributed to factors such as thermal contraction of the adjacent panels and the impact of freeze–thaw cycles, as well as the fact that cracks are more likely to propagate when the sealant product has a reduced modulus, which occurs at colder temperatures. The lower modulus induces greater stress along the sealant–substrate interface, leading to increased crack growth compared to warmer periods when the sealant has a higher modulus. The extension of sealants due to the contraction of the substrates at lower temperatures may induce stress on the material, potentially leading to the formation and propagation of cracks. Freeze–thaw cycles, common in cold seasons, can exacerbate this effect by causing the sealants to expand and contract repeatedly, placing additional stress on them and accelerating crack growth.

Conversely, during hot seasons, such as the first summer, the observed average crack growth rate of 2 × 10^−6^ mm·h^−1^ suggests a relatively lower propensity for crack development. Higher temperatures generally contribute to the lower extension of sealants, potentially mitigating the stresses that could lead to crack formation. However, it is important to consider other environmental factors, such as UV exposure and moisture levels, that can also influence the sealant’s behavior during hot seasons.

### 2.5. Climate Change Effect on Crack Growth

Anticipating how climate change may impact the service life of various construction materials is crucial for material scientists, engineers, and urban planners. This knowledge highlights the evolving challenges and opportunities in selecting materials that are suitable for a future characterized by changing climate conditions. 

Figure 14 offers a visualization of the crack growth in a hypothetical sealant installed in Estevan Point based on two distinct sets of climate data. The first part of the analysis utilizes real historical climate data for the years 1836 and 1936 to calculate the progression of cracks in the sealant until failure. In contrast, the second part of the analysis uses projected climate data for the years 2046, 2056, and 2066 to predict the crack growth in the sealant until failure is reached under a future climate scenario. Interestingly, the time-to-failure data, inset in the table within the figure, reveal a noticeable prolongation in the lifespan of the sealants under future climate scenarios. This extended durability can be primarily attributed to the increase in average annual temperatures, a direct consequence of ongoing climate change. 

This deceleration can be attributed to the profound impacts of global warming. The historical context offers a baseline for understanding how sealant integrity may evolve, while the forecasted future climate data, projecting a 2.5-degree Celsius increase in average temperature, highlight the critical role of environmental factors in shaping material performance. Higher average temperatures might result in reduced brittleness of the elastomeric sealants and possibly a slower rate of crack propagation since, generally, for this family of sealants, the flexural strength decreases with rising temperature [42]. However, it is important to note that while higher temperatures can delay the onset of cracking, the elevated temperature might also accelerate aging or degradation processes with prolonged exposure [43]. However, the prolonged service life calculated here is based on the constraints of the model, where temperature and relative humidity play a central role. This calculation does not account for other variables such as UV loads, which also need to be considered as they are important factors.

## 3. Conclusions

This study utilized historical climate data to assess how cracks evolve under diverse climatic conditions in Canada. This approach highlights the practical importance of considering long-term environmental trends when evaluating material durability and structural resilience. The comprehensive analysis of hypothetical silicone sealants installed across diverse locations in Canada highlighted the intricate interplay between climatic conditions and sealant performance. The crucial impact that installation temperature and temperature variation range have on sealant crack growth was also illustrated, and it was found that increasing installation temperature is key for accelerated simulations. Based on the analysis, historical assumptions in the existing models for joint failure overlook certain environmental stressors, which should be considered in the future. The scatter plots and climate classifications offered a systematic overview of how temperature and precipitation levels impact crack growth. The analysis of time-to-failure uncovered complex relationships between mean temperature, relative humidity, and sealant durability. The findings highlight the sealant’s sensitivity to seasonal temperature variations, with a notably greater impact on crack growth rates during colder seasons. The accelerated crack growth rate during colder periods was attributed to freeze–thaw cycles, as well as extreme extension of the sealants caused by drastic contraction of the substrates, which itself is linked to the high simulated installation temperature of 30 °C. The relatively lower likelihood of crack development in hot seasons was linked to lower levels of sealant extensions. An investigation applying projected climate data investigation underscored the significant influence of climate change on the durability of sealants, with higher temperatures potentially extending their lifespan. However, there is an ongoing need for a balanced consideration of prolonged exposure, which is not considered in the Shephard crack growth model, as it may accelerate aging processes and impact overall performance.

In summary, this manuscript provides valuable insights into the dynamic nature of sealant behavior in diverse climates, providing a foundation for refining models and optimizing sealant applications in real-world scenarios. The study’s results are crucial for designing and selecting sealants capable of withstanding varied environmental conditions. Furthermore, the findings are essential for implementing effective maintenance strategies specifically tailored to cold climates, such as those in Canada.

## Data Availability

The original contributions presented in the study are included in the article. Further inquiries can be directed to the corresponding author.

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
