# Peer review of "Crack Length of Elastomeric Sealants and Their Service Life in Contrasting Canadian Climates: Effects of Climate Change"

_polymers, 2024, doi:10.3390/polym16142039_

Round 1

Reviewer 1 Report

Comments and Suggestions for Authors

1.      The general factors affecting sealants to the specific application of the Shephard crack growth model should be included.

2.      What is the impact of environmental factors on the longevity of polymer-based sealants.

3.      What is the application of the Shephard crack growth model

4.      How the climatic data were classified into four categories and the criteria used for this classification.

5.      It mentions of 564 locations across Canada, provide geographical distribution.

6.      How these locations were chosen and how it comes under broader Canadian climate.

7.      The statement that colder climates tend to exhibit higher cumulative crack growth is not clear.

8.      Discuss on the interplay between climatic variation and material longevity.

9.      Explain the potential confounding factors that influence the crack growth beyond climatic variables.

10.  What is the relationship between the obtained findings and the polymer-based sealant longevity in construction.

Comments on the Quality of English Language

The linguistic level and the mechanics of English writing are not appropriate for publication. There are few grammatical and typing errors in the manuscript, so please check and revise. The way of writing is not clear and it is difficult for the readers to understand. The paper should be rewritten and proofread again thoroughly. Extensive editing of English language is required.

Author Response

Reviewer #1:

  1. The general factors affecting sealants to the specific application of the Shephard crack growth model should be included.

Authors’ response: Thank you for your insightful comment. Temperature and relative humidity are indeed already included in the Shephard crack growth model. Incorporating other environmental factors, such as UV exposure, wind load, and chemical environment requires extensive work. Currently, we are running a decade-long project on the outdoor aging of sealants, evaluating the factors influencing crack growth. The results from this project are essential for modifying the present models to include additional factors. Therefore, while we acknowledge your comment, addressing it is not possible at the moment. However, we want to inform you that we are already considering this and actively working on it. We appreciate your suggestion and are committed to addressing these factors in our ongoing research.

  1. What is the impact of environmental factors on the longevity of polymer-based sealants.

Authors’ response: Thank you for your question regarding the impact of environmental factors on the longevity of polymer-based sealants. The manuscript already addresses this in the following text (highlighted in the manuscript). This section of the manuscript highlights the significant impact of mean temperature and relative humidity on sealant longevity. It shows that sealants in areas with lower mean temperatures fail sooner due to increased stress and microcrack formation. Moreover, the combined influence of temperature and humidity indicates that higher relative humidity can also shorten the lifespan of sealants. This comprehensive analysis underscores the importance of considering both temperature and humidity in evaluating the durability of polymer-based sealants.

  1. What is the application of the Shephard crack growth model

Authors’ response: The manuscript already addresses this question in the text (highlighted in the manuscript) as below:

"The Shephard crack growth model is recognized for its application in investigating how cracks in sealant products behave under cyclic loading. This model was used to predict crack growth rate in sealants located in four contrasting climates in North America: Phoenix, Miami, Ottawa, and Winnipeg."

Additionally, we reference the document by Shepherd et al. (1995) Measuring and Predicting Sealant Adhesion, Ph.D. (Materials Engineering Science) Thesis, Virginia Polytechnic Institute and State University, Blacksburg, VA, USA. which serves as the basis for our assessment. The paper explains that this model also considers other variables, including installation temperature and the change of modulus over time. Based on this model, estimates of time to failure are made concerning specific failure criteria.

  1. How the climatic data were classified into four categories and the criteria used for this classification.

Authors’ response: Thank you for your question regarding the classification of climatic data. The climatic data were inherently from four distinct sources.

  1. Hourly Time Series: The first dataset consists of hourly time series of temperature and relative humidity from 564 monitoring locations in Canada, gathered from the Canadian Weather Energy and Engineering Datasets (CWEEDS) developed by Environment and Climate Change Canada.
  2. Historical Climate Estimates: The second dataset includes historical climate estimates spanning 1936-2015, prepared as part of the Twentieth Century Reanalysis Project version 3.
  3. Canadian Climate Normals: The third dataset comprises the Canadian Climate Normals, which represent 30-year averages to define and characterize the average climatic conditions of specific locations across Canada.
  4. Future Projected Climate: The fourth dataset is the future projected climate dataset, prepared for 564 locations widely distributed across Canada (Gaur et al. 2022).

Each of these datasets serves a different purpose in our analysis and provides a comprehensive view of climatic conditions relevant to our study.

  1. It mentions of 564 locations across Canada, provide geographical distribution.

Authors’ response: The geographical distribution of the 564 locations is already provided in Figure 1.

  1. How these locations were chosen and how it comes under broader Canadian climate.

Authors’ response: Thanks for the question. Based on the information gleaned from the scatter plot, 18 locations with the most contrasting climatic characteristics were selected. By first selecting the locations at the corners of the scatter plot, then around the peripheries, then filling in with a random representative selection from the interior of the scatter plot. These locations were chosen to ensure a comprehensive representation of the diverse climatic conditions across Canada. We aimed to capture the full spectrum of Canada's climate variability, from the extreme cold of northern regions to the milder conditions in southern areas, as well as the differences between coastal and inland climates. By selecting locations with the most contrasting climatic characteristics, we can more effectively analyze and understand how varying environmental factors impact the performance and longevity of building materials. This strategic selection of locations allows us to draw broader conclusions about the Canadian climate, ensuring that our findings are applicable across the country's diverse climatic regions.

  1. The statement that colder climates tend to exhibit higher cumulative crack growth is not clear.

Author’s response: Thank you for pointing that out. The statement has been clarified in the manuscript. We have revised the text to specify that colder climates with lower precipitation levels exhibit higher cumulative crack growth due to increased induced stress leading to more frequent microcrack formation and propagation, ultimately resulting in higher cumulative crack growth over time

  1. Discuss on the interplay between climatic variation and material longevity.

Author’s response: As per the reviewer’s comment, additional discussion has been incorporated into the relevant text to enhance clarity for readers. The longevity of sealants, as evidenced by their failure and crack growth rates, is deeply influenced by environmental stressors. Figure 11 in the manuscript presents time-to-failure data for sealants installed at Estevan Point over different decades. The observed variability in failure times closely correlates with the installation year, underscoring the pivotal role of environmental factors. This trend aligns with broader climate patterns since the mid-20th century due to global warming.

  1. Explain the potential confounding factors that influence the crack growth beyond climatic variables.

Authors’ response: It is quite challenging to answer this question comprehensively since involves considering numerous factors that can influence the crack growth of sealants. These factors include, but are not limited to:

  • Material manufacturing issues such as defects or impurities. Variations in sealant formulation and composition. These may affect the sealant's flexibility and overall performance.
  • Installation conditions and techniques including surface preparation, joint design, and curing conditions.
  • Exposure to chemicals and liquids
  • Mechanical stresses resulting from inappropriate structural installation that cause unplanned structural movements and vibrations
  • Change in the surface of the applied substrate due to aging, degradation, or weathering

However, reaching a solid conclusion on how these factors can affect the longevity of sealants by influencing crack initiation and propagation requires extensive study which is out of the scope of our current study.

  1. What is the relationship between the obtained findings and the polymer-based sealant longevity in construction.

Authors’ response: The current study elucidates how climatic variations impact the longevity of polymer-based sealants in construction by investigating how environmental stressors influence the crack growth and ultimate durability of sealants. By analyzing historical climate data from diverse Canadian locations, we found that colder climates with lower precipitation levels are expected to experience higher crack growth rates in sealants due to thermal cycling and freeze-thaw cycles. Seasonal variations also play a significant role, with temperature fluctuations influencing sealant performance differently across different seasons. Our findings highlighted the need for tailored material selection in varying environmental conditions.

Comments on the Quality of English Language

The linguistic level and the mechanics of English writing are not appropriate for publication. There are few grammatical and typing errors in the manuscript, so please check and revise. The way of writing is not clear and it is difficult for the readers to understand. The paper should be rewritten and proofread again thoroughly. Extensive editing of English language is required.

Authors’ response: Thank you for your valuable feedback. We have thoroughly reviewed and revised the manuscript to improve its clarity, flow, and overall readability. The updated manuscript reflects these improvements in red.

Reviewer 2 Report

Comments and Suggestions for Authors

The manuscript is oriented to present mainly very broad source data on climate change in the area. They are interesting information. However, in the context of this special issue, I think there is too little information about polymers (as construction materials). It is impossible to distinguish the research part here. Although the information presented is interesting, I am not convinced of the scientific level of the manuscript. I make very minor comments below.

-Entering the Highlights part is not necessary.

-lines 116, 184, 225, 232, 251, 268, 284, 287, 310, 352, no literature cited

- The way the equations are written does not comply with the requirements of the journal

-line 211- standardize the way literature is served

I think that conditionally it can be accepted for publication if there is approval from the other reviewers and editors.

Author Response

Reviewer #2:

The manuscript is oriented to present mainly very broad source data on climate change in the area. They are interesting information. However, in the context of this special issue, I think there is too little information about polymers (as construction materials). It is impossible to distinguish the research part here. Although the information presented is interesting, I am not convinced of the scientific level of the manuscript. I make very minor comments below.

Authors’ response: Thanks for the reviewer’s thoughtful feedback. We understand your point regarding the emphasis on climate change data. However, we believe our manuscript aligns well with the scope of the special issue on Sustainable Polymeric Materials in Building and Construction. Our research addresses the following key topics of the special issue:

  • Resilience in building practice: Our work explores the impact of climate change on the resilience of elastomeric sealants.
  • Real-world performance and life cycle assessment: Our work provides data on sealant performance and service life under varying climates.
  • Durability and service life prediction: Our work focuses on predicting the service life and durability of sealants in different conditions.
  • Integration in sustainable practices: Our work supports the use of durable sealants in sustainable building practices.

The authors hope that the above points clarify how our manuscript fits within the special issue's scope.

- Entering the Highlights part is not necessary.

Authors’ response: The authors believe that including “Highlights” is beneficial as they provide a concise summary of the key findings. This can help readers quickly grasp the significance of the work, therefore, the authors would like to retain the Highlights section to enhance the accessibility and visibility of our research findings.

- lines 116, 184, 225, 232, 251, 268, 284, 287, 310, 352, no literature cited

Authors’ response: Thanks for the feedback. We have added citations to some relevant sections as suggested. However, most of the statements are based on the authors' own expertise and findings, thereby do not require citations.

- The way the equations are written does not comply with the requirements of the journal

Authors’ response: According to the “Instructions for Authors” the requirement for the equations is given as below:

  • Equations: If you are using Word, please use either the Microsoft Equation Editor or the MathType add-on. Equations should be editable by the editorial office and not appear in a picture format.

The equations in the manuscript comply the above requirement.

- line 211- standardize the way literature is served

Authors’ response: Revised as per comment.

I think that conditionally it can be accepted for publication if there is approval from the other reviewers and editors.

Authors’ response: The authors appreciate your minor comments to further enhance the scientific rigor and clarity of our work. We also tried to address your concerns about the alignment with the scope of the special issue on Sustainable Polymeric Materials in Building and Construction. As mentioned above, our study is directly relevant to several key topics outlined in the scope. It explores the impact of climate change on the resilience of elastomeric sealants, which are crucial for maintaining the integrity of building envelopes. We provide extensive data on the performance and service life of sealants under various Canadian climates, reflecting real-world conditions. Our focus is on predicting the durability and service life of sealants in different climatic conditions, which is essential for long-term planning and maintenance. By highlighting the importance of durable sealants, our study supports their integration into sustainable building practices, promoting long-term environmental and economic benefits. We hope this explanation reassures you of our study's relevance to the scope of the special issue and look forward to the approval from the other reviewers and editors.

Reviewer 3 Report

Comments and Suggestions for Authors

(reviewer's comments in blue)

1. What is the main question addressed by the research?

The main question addressed is the testing of the service life of elatomeric sealants under the extreme conditions of low temperature encountered in Canada.
This is fundamental not only tpo a specific Canada applications but to all countries where both extrmely low temperature conditions are encountered and even more where wide fluctuations of temperatures between maximum and minimumum are recorded. This is true as elastomericsealants harden and lose their elasticity, there is their reason of existing, once these temperature conditions are encountered. In this the paper is very worthwhile and is likely to be widely red.

2. What parts do you consider original or relevant for the field? What specific gap in the field does the paper address? What does it add to the subject area compared with other published material?

There is not in depth published material on this subject, especially, and this is crucial, on field expoeriments in many fields locations. In fact the majority of previous paper address testing of elongation under standard operating conditions, but under laboratory conditions, hence accelerated lab tests, that unfortunately have a very limited significance for the real operational situation in the field of these materials.
without really addressing or modeling the extreme conditions. This is the originality of this paper.

3. Illustrate what are, in your opinion, its strengths and weaknesses (this is an essential step, as the editor will consider the reasoning behind your recommendation and needs to understand it properly);

Considering the novelty,the reasons from this reviewer side are what already exposed in points 1 and 2 above

4. What specific improvements should the authors consider regarding the methodology? What further controls should be considered?

What should be addressed and would add to the publication would be for the authors to produce a reliable model of durability considering all the parameters and data collected in the numerous field tests. While this would be of considerablke utility it is preposterous for a reviewer to demande such an extensive work in a paper that I am sure is just the initial one leading eventually to a second paoer presenting such model of applied operational interest. I would leave this paper exactly as it is.

5. Please describe how the conclusions are or are not consistent with the evidence and arguments presented. Please also indicate if all main questions posed were addressed and by which specific experiments.

Please see my remarks in point 4 above.

6. Please include any additional comments on the tables and figures and the quality of the data.

There are no additional remarks. The paper is indeed outstanding and what required more in point 4 above should be addressed in a second paper rather than be included in thius one

Author Response

Reviewer #3:

  1. What is the main question addressed by the research?

The main question addressed is the testing of the service life of elastomeric sealants under the extreme conditions of low temperatures encountered in Canada. This is fundamental not only to specific Canada applications but to all countries where both extremely low-temperature conditions are encountered and even more where wide fluctuations of temperatures between maximum and minimum are recorded. This is true as elastomeric sealants harden and lose their elasticity, there is their reason for existing, once these temperature conditions are encountered. In this, the paper is very worthwhile and is likely to be widely read.

Authors’ response: The authors are very grateful to hear the reviewer’s positive overall opinion and would like to thank the reviewer for the valuable comments that helped us to improve the quality of our manuscript.

  1. What parts do you consider original or relevant for the field? What specific gap in the field does the paper address? What does it add to the subject area compared with other published material?

    There is not in-depth published material on this subject, especially, and this is crucial, on field experiments in many field locations. In fact the majority of previous papers address testing of elongation under standard operating conditions, but under laboratory conditions, hence accelerated lab tests, that unfortunately have a very limited significance for the real operational situation in the field of these materials. without really addressing or modeling the extreme conditions. This is the originality of this paper.

Authors’ response: We acknowledge the limited published literature, particularly on field experiments conducted across multiple locations, which are crucial for assessing the real-world conditions of these materials. As part of a comprehensive long-term project on outdoor aging, we are validating and refining our models to accurately simulate and predict sealant behavior under diverse environmental stresses, including extreme conditions. Your feedback confirms that our approach to addressing these challenges is aligned with the needs of advancing the field.

  1. Illustrate what are, in your opinion, its strengths and weaknesses (this is an essential step, as the editor will consider the reasoning behind your recommendation and needs to understand it properly);

    Considering the novelty,the reasons from this reviewer side are what already exposed in points 1 and 2 above

Authors’ response: Thank you for recognizing the novelty of our manuscript.

  1. What specific improvements should the authors consider regarding the methodology? What further controls should be considered?

What should be addressed and would add to the publication would be for the authors to produce a reliable model of durability considering all the parameters and data collected in the numerous field tests. While this would be of considerable utility it is preposterous for a reviewer to demand such extensive work in a paper that I am sure is just the initial one leading eventually to a second paper presenting such a model of applied operational interest. I would leave this paper exactly as it is.

Authors’ response: We acknowledge and appreciate the reviewer's comment, which is in line with Reviewer #1's initial feedback too. We are fully aware of the current limitations of the model in providing comprehensive and reliable results. As mentioned previously, our ongoing decade-long outdoor aging project is dedicated to compiling climatic data and stress factors to incorporate additional environmental variables and refine existing models. While fully addressing these aspects is not feasible within the scope of this paper, we are actively pursuing these enhancements. We value the reviewers' insights and remain committed to addressing these factors in our ongoing research endeavors.

  1. Please describe how the conclusions are or are not consistent with the evidence and arguments presented. Please also indicate if all main questions posed were addressed and by which specific experiments.

    Please see my remarks in point 4 above.

Authors’ response: We appreciate the reviewer's thoughtful comment. Our conclusions align with the evidence and arguments presented in the manuscript. We have addressed the main questions focusing on crack growth and time-to-failure analysis across diverse climatic conditions in Canada using historical and projected data, supported by the Shephard crack growth model. The current paper focuses on laying the groundwork and highlighting the need for a reliable durability model encompassing all parameters from extensive field tests. The suggestion to develop a comprehensive model incorporating all parameters and field test data is indeed valuable and aligns with our long-term research goals as mentioned above.

  1. Please include any additional comments on the tables and figures and the quality of the data.

    There are no additional remarks. The paper is indeed outstanding and what required more in point 4 above should be addressed in a second paper rather than be included in this one

Authors’ response: It is indeed our intention to advance towards such a model in subsequent papers. We appreciate your understanding of the scope and progression of our research.

Round 2

Reviewer 1 Report

Comments and Suggestions for Authors

I have carefully reviewed the revised manuscript. I appreciate the thoroughness of research and the clarity of presentation. I am pleased to note that the revisions made in response to the initial comments have significantly strengthened the manuscript. The clarity of the methodology, the presentation of results, and the discussion of findings have all been notably improved. Given the high quality of the revisions, I am confident that the manuscript is now suitable for publication in its current form. I appreciate your efforts in addressing the comments and congratulate you on a well-executed study.